# Sea Level Variability in the Swedish Exclusive Economic Zone and adjacent seawaters: Influence on a Point Absorbing Wave Energy Converter

Valeria Castellucci[1] and Erland Strömstedt[1]

[1] Div. of Electricity, Dept. of Engineering Sciences, Ångström Laboratory, Uppsala University, Box 534, 75121, Uppsala, Sweden

*Correspondence to*: Valeria Castellucci (valeria.castellucci@angstrom.uu.se)

**Abstract.** Low-frequency sea level variability can be a critical factor for several wave energy converter (WEC) systems, for instance linear systems with a limited stroke length. Consequently, when investigating suitable areas for deployment of those

WEC systems, sea level variability should be taken into account. In order to facilitate wave energy developers in finding the most suitable areas for wave energy park installations, this paper describes a study that gives them additional information by exploring the annual and monthly variability of the sea level in the Baltic Sea and adjacent seawaters, with focus on the Swedish Exclusive Economic Zone. Ten years of reanalysis data from the Copernicus project have been used to conduct this investigation. The results are presented by means of maps showing the maximum range and the standard deviation of the sea

level with a horizontal spatial resolution of about 1 km. A case study illustrates how the results can be used by the WEC developers to limit the energy absorption loss of their devices due to sea level variation. Depending on the WEC technology one wants to examine, the results lead to different conclusions. For the Uppsala point absorber L12 and the sea state considered in the case study, the most suitable sites where to deploy WEC parks from a sea level variation viewpoint are found in the Gotland Basins and in the Bothnian Sea, where the energy loss due to sea level variations is negligible.

**Nomenclature**

| | |
|---|---|
| $H_s$ | Significant wave height |
| MSSHR | Maximum sea surface height range |
| $MSSHR_y$ | Annual maximum sea surface height range based on $SSH_{1h}$ |
| $MSSHR_{10y}$ | Decadal maximum sea surface height range based on $SSH_{1h}$ |
| $MSSHR_{m,10y}$ | Monthly maximum sea surface height range for each month averaged over 10 years, based on $SSH_{1h}$ |
| SD | Standard deviation |
| $SD_y$ | Annual standard deviation of $SSH_{1h}$ |
| $SD_{10y}$ | Decadal standard deviation of $SSH_{1h}$ |
| $SD_{m,10y}$ | Monthly standard deviation of $SSH_{1h}$ for each month, pooled over 10 years |
| $SDR_{10y}$ | Standard deviation of the $MSSHR_y$ over 10 years |

| SEEZ | Swedish Exclusive Economic Zone |
|------|----------------------------------|
| SL | Sea level |
| SMHI | Swedish Meteorological and Hydrological Institute |
| SSH | Sea surface height |
| $SSH_{1h}$ | Sea surface height with hourly resolution |
| SWERM | Swedish wave energy resource mapping |
| $T_e$ | Energy period |

## 1 Introduction

In the Baltic Sea, the variations of sea level (SL) are controlled by meteorological and climatological processes, including the hydrological balance (Johansson et al., 2001). Tides give a small contribution to these variations, since the Scandinavian basins

are characterized by low tidal levels during the year. As suggested by (Ekman, 2009), the Baltic Sea has no real tides, but storm winds could raise the sea level locally by more than 2.4 m. The largest amplitudes reach up to $3 - 4$ m as storm surges and seiches in the Gulf of Finland (Kulikov et al., 2014). In general, the tide is a few centimeters high, with peaks of about 24 cm in the Gulf of Finland, as estimated by (Medvedev et al., 2016). In (Samuelsson and Stigebrandt, 1996) the sea level variations are classified as 'external' and 'internal': respectively, long-term winds transporting water between the Atlantic

Ocean and the Baltic Sea, and short-term winds together with changes of density and barometric pressure, redistributing water within the Baltic Sea. Those two types of variability may exhaustively explain the low-frequency SL changes in the Baltic Sea. Being that those changes are predominantly influenced by air pressure and wind stress, the variability is mostly of random character and seasonal cycles are dominant (Kulikov et al., 2014). According to Hünike et al. (2005) during the summer, temperature and precipitation explain part of the SL variability except in the Kattegat region. Furthermore, SL exhibits an

annual cycle peaking in the winter months.

SL variations are of great importance and have been thoroughly investigated by many researchers for example with the purpose of broadening the knowledge on climate change (IPCC, 2018), spatial patterns (Ekman, 1996) (Donner et al., 2012), land uplift (Miettinen et al., 1999), and the pole tide (Ekman, 1996) (Medvedev et al., 2014) in the Baltic Sea. The reason why the study presented in this paper has been carried out is to give wave energy developers additional information to use when looking for

suitable sites for their devices. Generically, a wave energy converter (WEC) extracts energy from high-frequency waves, while it might be negatively affected by low-frequency SL changes depending on its design. The Uppsala WEC, shown in Fig. 1, is considered as an example. The WEC consists of a surface-floating buoy vertically driving an encapsulated linear generator on top of a foundation acting as a fixed reference on the sea floor. The tension in the connection line and the distance between the buoy and the sea bed is influenced by low-frequency SL variations: for a significantly low SL, the connection line is slack and

the translator rests on the bottom of the generator; while for a significantly high SL, the translator continuously hits the upper end-stop, which results in additional stresses on the hull of the generator and in a reduced stroke of the translator itself. In both cases, the energy absorption decreases drastically, together with the lifetime and survivability of the WEC (Castellucci et al.,

2016). The same problem is experienced by other technologies, such as oscillating water columns, as suggested by Muetze and Vining (2006) and by Lopez et al. (2015), and in more general terms by WECs which have a part that is fixed in position relative to the seabed and a part that moves with the waves. Well-known point absorbers, such as Carnergie CETO (Kenny, 2014), Ocean Power Technologies Powerbuoy (OPT, 2018), and Archimedes Wave Swing (Beirdol et al., 2007) are challenged by SL changes, either because of a limited stroke length or because of the exponential decrease in available energy with depth. The work presented in this paper is part of a bigger wave energy project on Swedish wave energy resource mapping (SWERM) financed by the Swedish Energy Agency (Strömstedt et al., 2017). The project aims to generate and combine different layers of information, like bathymetry, sea ice coverage, wave climate, wave energy conversion potential, etc., for the Swedish Exclusive Economic Zone (SEEZ) in order to identify the most suitable areas for wave energy conversion. Within this framework, the study here conducted aims to evaluate the SL information layers: the paper presents the results for the SL variations over a larger area, that includes the SEEZ and adjacent seawaters (see Fig. 2). The input data and the methodology are discussed in Chapter 2. The results are shown in Chapter 3 by means of maps. GIS layers will be available on-line or on request at the end of the project, so that detailed data can be extracted. Finally, discussion and conclusion are presented in Chapters 4 and 5.

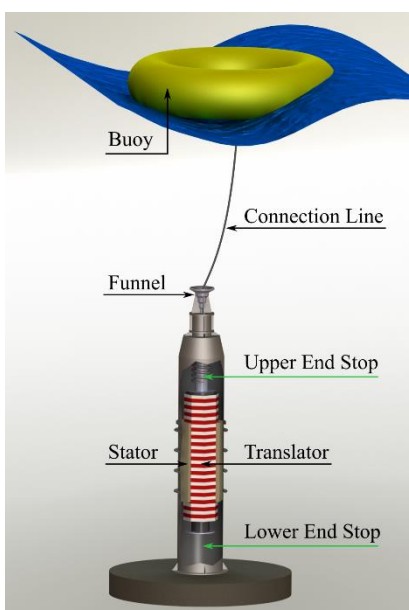

**Figure 1: Illustration of the point absorber WEC developed at Uppsala University. Reprinted from Castellucci et al. (2016).**

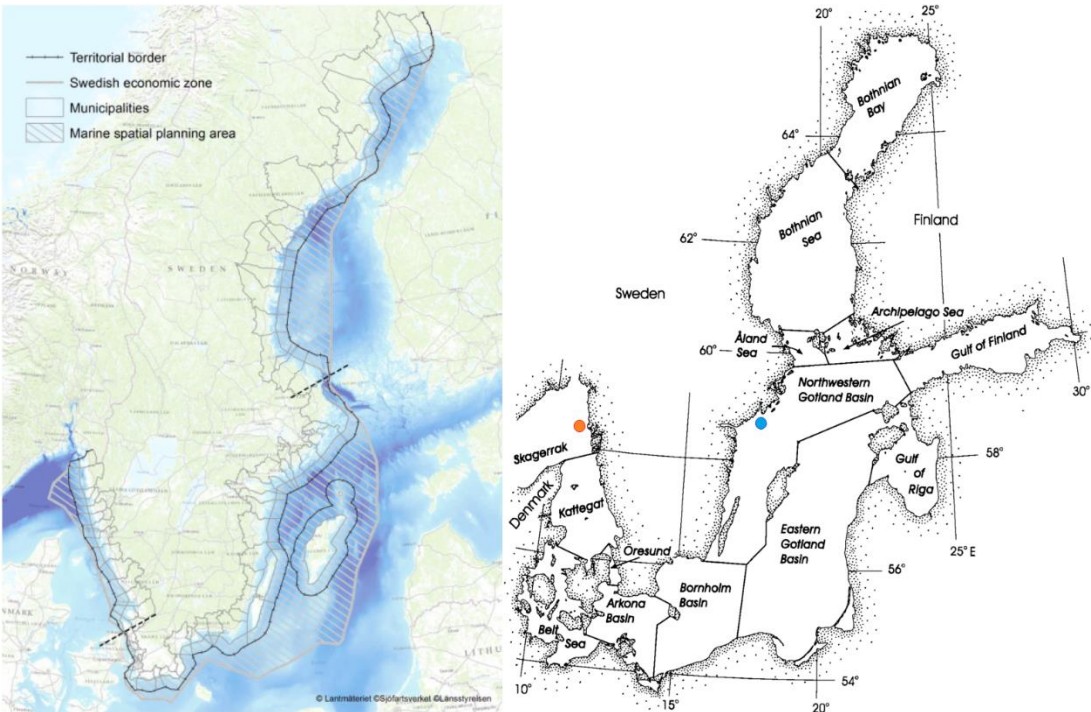

**Figure 2: Left) Map of the SEEZ around Sweden in focus for this study. Right) Map of the considered water basins. The same basin terminology is used throughout the article. Credits to HELCOM (2018). The blue marker indicates the station at Landsort, while the orange marker points at the station of Väderöarna.**

## 2 Data and methods

In order to produce comprehensive maps of sea surface height (SSH) in the Baltic Sea as a whole, it is necessary to interpolate the available data over space and time. However, measurement stations are located far from each other, even more than 100 km, and some are visited only once a month. Some may lack observations for very long time periods. In order to compensate for those deficiencies, observations are combined with model simulations to obtain a homogeneous data set with high resolution in time and space, and reasonably close to observations. This can be achieved with a process called data assimilation, in which observations are used to update the circulation model to keep it from deviating too far away from reality (Axell and Liu, 2016).

The circulation model used by the Swedish Meteorological and Hydrological Institute (SMHI) to produce the reanalysis data used in this study is HIROMB (High-Resolution Operational Model for the Baltic). HIROMB has open boundaries in the western English Channel and in the northern North Sea. For SSH, HIROMB uses data from the coarse storm-surge model NOAMOD (44 km resolution), whereas climatological monthly mean values are used for salinity and temperature. Moreover, ice variables are assumed to be zero at the boundary. The meteorological forcing is from the High-Resolution Limited Area

Model (HIRLAM, 2019), with a resolution of 22 to 11 km. The chosen data assimilation method is the 3DEnVar (3-D Ensemble Variational) data assimilation, a multivariate method where many variables are affected by each observation. The observations assimilated into this model are ice concentration, level ice thickness, sea surface temperature, and profiles of salinity and temperature. The directly affected model variables are the same, i.e. ice concentration, level ice thickness, salinity

and temperature. Other variables are affected indirectly to a small degree, including e.g. currents and SSH (through its effects on density). However, the differences in currents and SSH compared to a free run without data assimilation is rather small. For more information regarding the model description and validation see (Axell and Liu, 2016) and the product documentation (Copernicus, 2018). In general, the results obtained for SSH in the SEEZ and the adjacent seawaters are rather good: mean correlations of about 0.91 and mean RMS errors of about 9 cm are calculated by comparing hourly instantaneous model data

with corresponding coastal observations for three different years. The SSH data available on-line at *marine.copernicus.eu* have a spatial resolution of 1/20 degrees in the north-south direction and 1/12 degrees in the east-west direction, which translates into about 5.5 km resolution. The requirement set by the SWERM project is to work on a common grid of about 1 km$^2$, hence, the reanalysis data have been linearly interpolated with the purpose of fitting this grid. Moreover, a 10-year data set (2007 to 2016) with a temporal resolution of one hour has been chosen in order to examine the annual and monthly variability of the

$SSH_{1h}$ oscillations, neglecting extreme events. Within this study, the terms SL and SSH are generally interchangeable, while $SSH_{1h}$ refers more strictly to the data used to carry out the analysis. Fig. 3 shows an excerpt of the simulated model data from January 2014 to December 2015 at two representative locations: Väderöarna and Landsort, in the Skagerrak (Latitud: 58.5760, longitude: 11.0661) and in the Northwestern Gotland Basin (Latitud: 58.7404, longitude: 17.8655) respectively.

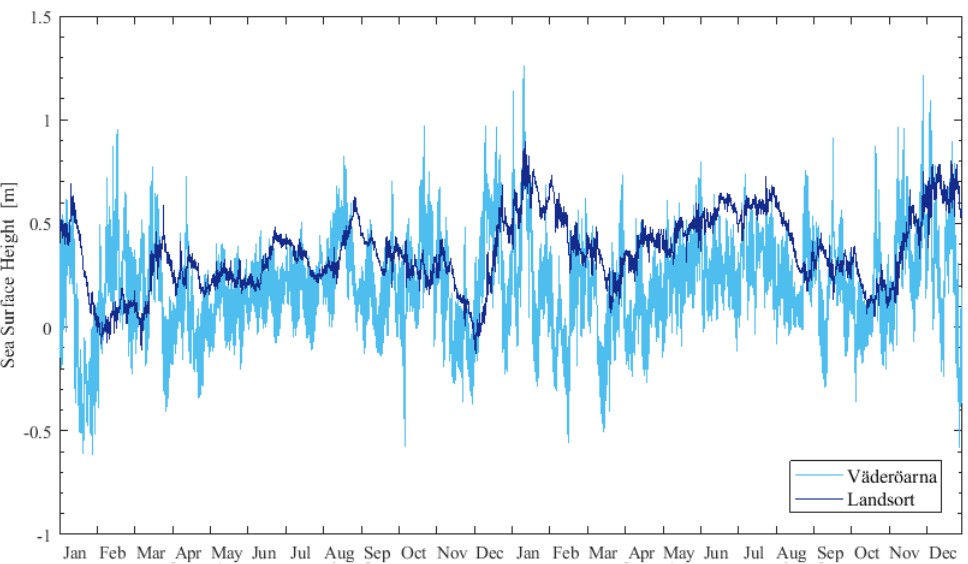

**Figure 3: SSH time series from January 2014 to December 2015 at the stations of Väderöarna in the Skagerrak and Landsort in the Northwestern Gotland basin.**

The metrics considered relevant to this study are the maximum range and the standard deviation of the SL variations. Note that both metrics are independent of the choice of reference level. The range, calculated as the difference between the highest $SSH_{1h}$ and the lowest $SSH_{1h}$ during the selected time period, gives an indication of the maximum variation of the SL. Some WEC technologies may be unaffected by variations below a certain range, like the Uppsala WEC in mild wave climates, as

discussed in Chapter 4. Furthermore, the highest absorption loss for a device can be estimated by WEC developers as presented in the case study in Chapter 3, and mitigation measures can be adopted. The standard deviation (SD), calculated as the square root of the variance for the chosen data set, quantifies the dispersion of the data from their mean value. The higher the SD, the more spread out the data points are from the expected value, hence, it is a measure of the variability of the SL variations. When selecting a site for WEC deployment, one may find it preferable to choose an area with as constant conditions as possible: the

frequency of occurrence of high ranges is greater for higher values of SD and the design costs for a WEC may increase with it. In general, the lower the standard deviation, the better it is. Moreover, both metrics, range and SD, are independent of the choice of reference level, which for SL is not always self-evident (Johansson et al., 2001). In fact, the data set provided by Copernicus have a zero mean value at the outer boundary, in the Atlantic. In the Baltic Sea, the SL is higher due to the density difference between the Atlantic Ocean and the Baltic Sea.

The SL range is calculated in Eq. (1) and (2) as the difference between the absolute maximum and minimum values over the 10-year data set of $SSH_{1h}$, denoted as $MSSHR_{10y}$, and over 10 years per each month, denoted as $MSSHR_{m,10y}$. In other words:

$$MSSHR_{10y} = max(SSH_{1h,i}) - min(SSH_{1h,i}) \qquad (1)$$

$$MSSHR_{m,10y} = max(SSH_{1h,m|_{10y}}) - min(SSH_{1h,m|_{10y}}) \qquad (2)$$

where $i = 1, 2 \dots N$ with $N$ being the number of all the $SSH_{1h}$ in the 10-year data set, and $m$ corresponds to the month of the year.

The SD has been obtained, using Eqs. (3) – (6), as the average of annual SDs over the 10-year data set, $SD_{10y}$, and as the square

root of the pooled variance to aggregate monthly SD over 10 years, $SD_{m,10y}$. More specifically:

$$SD_{m,y} = \sqrt{\frac{1}{n_{m,y} - 1} \sum_{j=1}^{n_{m,y}} \left[ SSH_{1h,j} - \overline{SSH_{1h,J}} \right]^2} \qquad (3)$$

$$SD_y = \sqrt{\frac{\sum_{m=1}^{12}(n_{m,y} - 1)SD_{m,y}^2}{\sum_{m=1}^{12}(n_{m,y} - 1)}} = \sqrt{\frac{(n_{1,y} - 1)SD_{1,y}^2 + (n_{2,y} - 1)SD_{2,y}^2 + \dots + (n_{12,y} - 1)SD_{12,y}^2}{(n_{1,y-1}) + (n_{2,y} - 1) + \dots + (n_{12,y} - 1)}} \qquad (4)$$

$$SD_{m,10y} = \frac{1}{10} \sum_{y=1}^{10} SD_{m,y} \qquad (5)$$

$$SD_{10y} = \frac{1}{10} \sum_{y=1}^{10} SD_y \qquad (6)$$

where $j = 1, 2, \dots n_{m,y}$ with $n_{m,y}$ equal to the number of $SSH_{1h}$ in a month ($m$) for the year ($y$), which may vary depending on the month and year, for the entire 10-year data set. The pooled variance in Eq. (4) is weighted taking into consideration that every month has a different number of days, hence, number of $SSH_{1h}$ values.

Finally, a case study is presented in order to give an idea of how the results can be used by wave energy developers. The
Uppsala WEC technology is considered. In particular the energy absorption of an L12 generator is simulated by hydrodynamic modelling. The following features are assumed: a cylindrical buoy of radius 3 m and draft 0.6 m; a translator stroke length of about 2.5 m; a total weight of the moving parts except the buoy of 10 tonnes; a damping factor of about 135 kNs/m. For more details regarding the model and its limitations see (Castellucci et al., 2016). For the mere purpose of providing an example of WEC energy absorption at different SLs, a sea state characterized by a significant wave height $H_s = 1$ m and energy period $T_e$
$= 5$ s is used as input to the model. These values are considered to be a reasonable approximation of the wave climate in the Baltic Sea (Soomere et al., 2007) (Soomere et al., 2011) (I. Zaitseva, 2013).

## 3 Results

The results for SL range and SD are summarized in Section 3.1.1 and 3.1.2 respectively. The energy absorption as a function
of the SL for an Uppsala WEC is estimated for a specific sea state and presented in Section 3.2.

## 3.1 Sea level metrics

### 3.1.1 Range

The MSSHR variations during the years 2007 to 2016 has been calculated from the interpolated reanalysis data sets. Figure 4 shows the highest monthly ranges over the 10-year period ($MSSHR_{m,10y}$) in the Scandinavian basins. Figure 5 shows, on the
left, the average of the annual maximum ranges ($MSSHR_y$) and, on the right, the absolut maximum range over 10 years ($MSSHR_{10y}$). The variability of $MSSHR_y$, estimated as the standard deviation of the $MSSHR_y$ over 10 years ($SDR_{10y}$), has a minimum value of 0.05 m between the Danish islands and the coast of Germany and a maximum of 0.5 m in the innermost part of the Gulf of Finland. In general, a quite moderate variation ($SDR_{10y} < 0.3$ m) is calculated along the Swedish coast. The time period from April to September (summer-time) appears to be the one with the lowest ranges compared to the period
October to March (winter-time) as shown in Fig. 4. The spatial pattern is clear and almost independent of the time of the year: the greatest oscillations of $MSSHR_{m,10y}$ occur in the Bothnian Bay, the Gulf of Finland, the Kattegat and in the Danish straits.

The legend in Fig. 4 is capped at 2 m to better illustrate the variations inside the SEEZ, but the SL can actually reach 4 m in the eastern parts of the Finnish gulf. The Northwestern Gotland Basin is the most stable area, characterized by $MSSHR_{10y}$ ranges of 1.2 to 1.5 m (see Fig. 5). However, during summer-time the range is likely to be lower than 0.7 m.

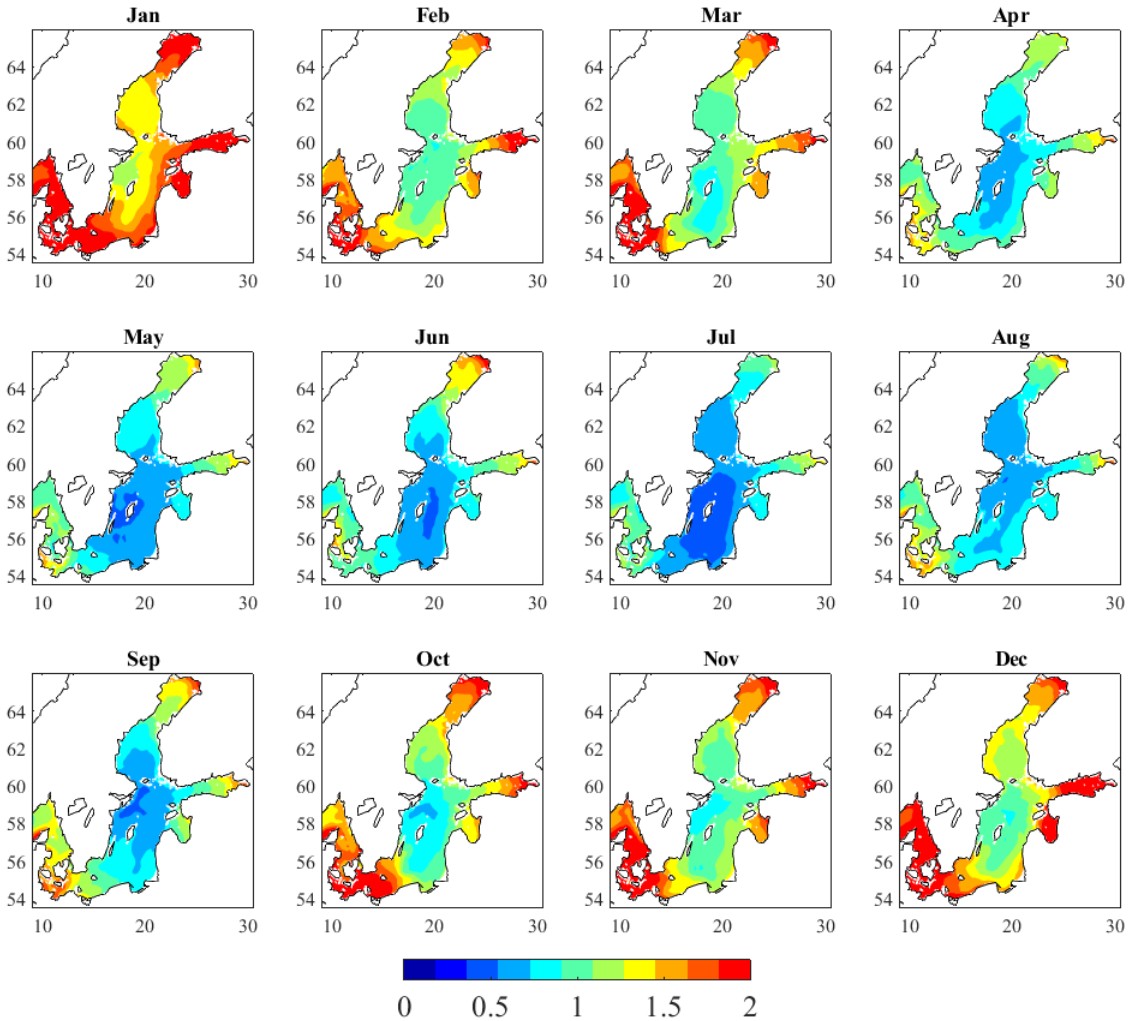

**Figure 4: $MSSHR_{m,10y}$ (Monthly maximum ranges [m] for each month over 10 years, 2007-2016, of re-analysis data). The red areas illustrate MSSHRs higher than about 1.8 m, up to 4 m.**

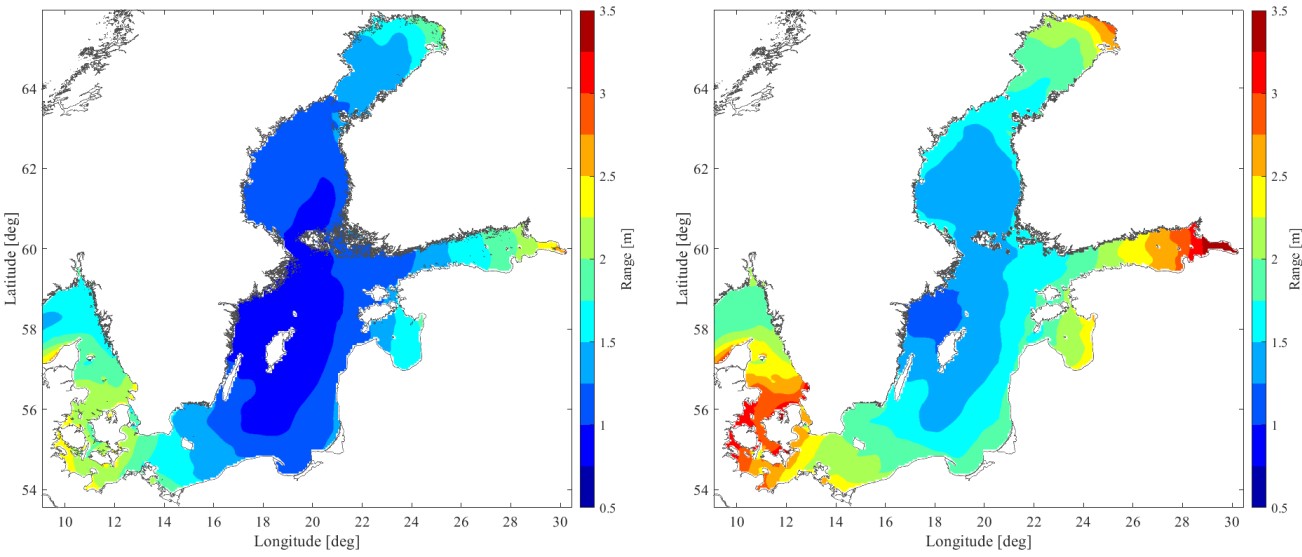

**Figure 5: Left) Average MSSHR$_y$ (average annual maximum ranges over the 10-year window). Right) MSSHR$_{10y}$ (decadal maximum ranges over the 10-year window). The colour scale is different from the one in Fig. 4 for ease of readability and visualization.**

### 3.1.2 Standard deviation

The standard deviation (SD) of the SSH$_{1h}$ has been evaluated in order to have a better understanding of the variability of the data set. The variance of the SSH$_{1h}$ has been calculated for each month according to Eq. (3) and, then, aggregated by month and averaged over the 10-year windows by computing a pooled SD using Eq. (4) and (5) in order to obtain SD$_{m,10y}$. The results are shown in Fig. 6. The average of the 10 annual SDs (SD$_{10y}$), calculated according to Eq. (6), is shown in Fig. 7.

With reference to Fig. 6, the spatial and temporal patterns are once again clear. In the Gotland Basins the pooled SD$_{m,10y}$ is the lowest, expecially in the summer-time when the SD$_{m,10y}$ values can be as low as 0.05 m (May). The SD$_{m,10y}$ increases as we move out from the center of the Baltic Sea and a peak of 0.4 m is calculated in the Skagerrak, by the northern coast of Danmark, during the month of January. In the same area, the SD$_{10y}$ is found to be 0.32 m, while the lowest SD$_{10y}$, about 0.08 m, is found in the Northwestern Gotland Basin (see Fig. 7). As expected, the variability of the data determined as the average of annual SD, SD$_{10y}$, turns out to have a smaller interval than the pooled monthly SD (SD$_{m,10y}$) used to aggregate monthly SDs over 10 years.

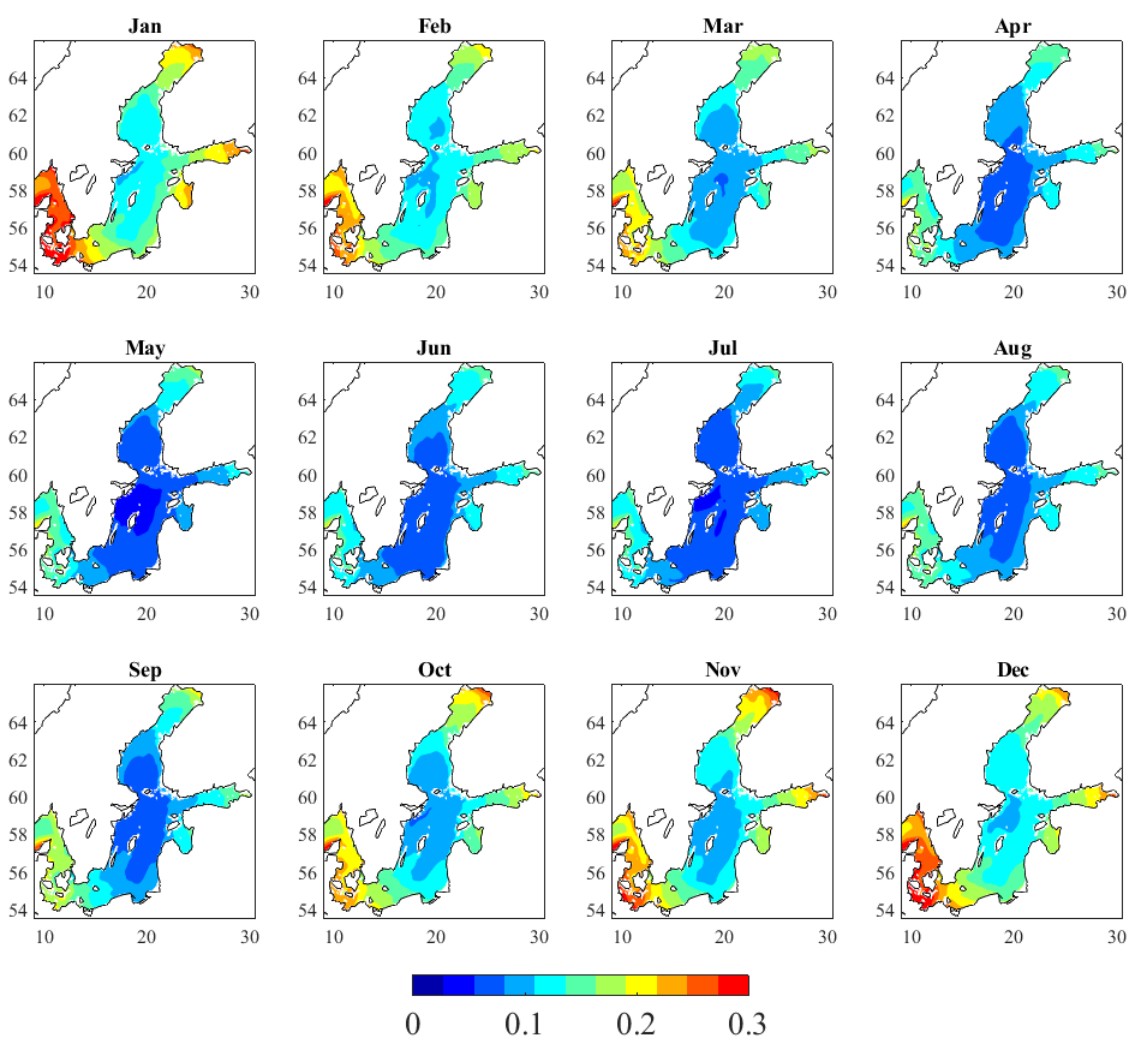

**Figure 6: SD$_{m,10y}$ (Monthly SD [m] for each month over 10 years, 2007-2016, of re-analysis data).**

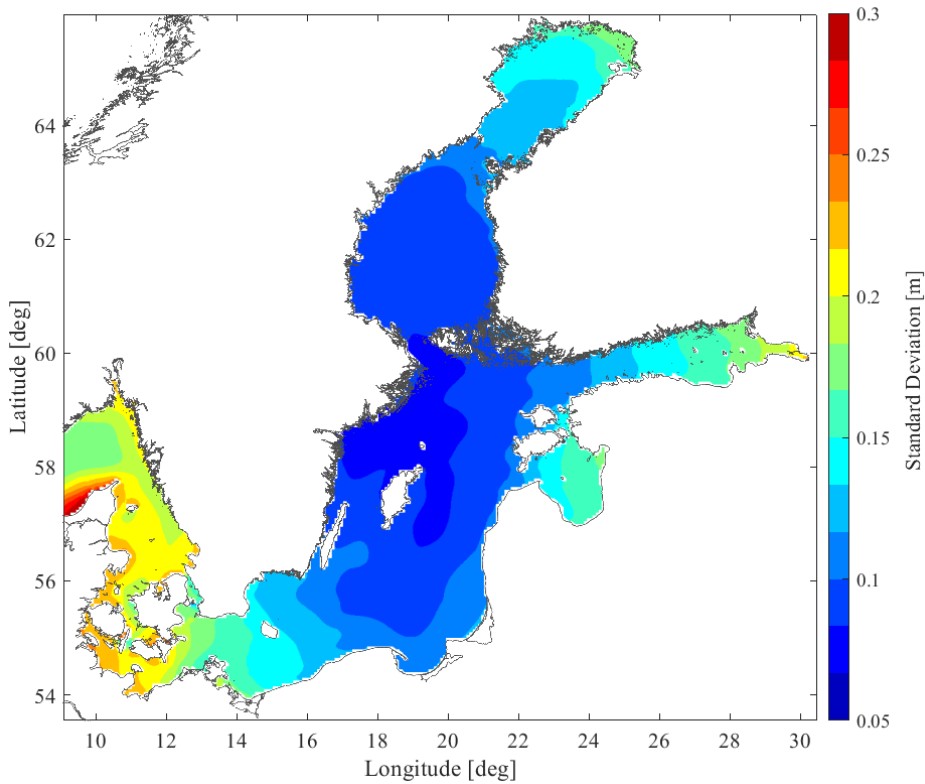

**Figure 7: SD$_{10y}$ (Decadal SD of the SSH$_{1h}$ over the 10-year window). The colour scale is different from the one in Fig. 6 for ease of readability and visualization.**

### 3.2 Case study

5  In Castellucci et al. (2016) the hydro-mechanic model that analyses the behaviour of a point absorber is described. In particular, the model evaluates how SL variations influence the power absorption, hence the energy production, of the Uppsala WEC across a representative scatter of wave climates. Note that power is absorbed as long as the translator moves within the stator (see Fig. 1). An example is presented in Fig. 8 with the purpose of pointing out the effect of SL changes on the performance of the Uppsala WEC denoted L12 (Castellucci et al., 2016). Let's assume that the hypothetical wave energy developer is

10  interested to deploy a wave energy park where the significant wave height is not greater than 1 m. The normalized annual energy absorption for different SL in the range of ±0.8 m is close to 100 % and it drops drastically for |SL| > 0.8 m, as illustrated in Fig. 8. When the SL exceeds the stroke length of the translator, then the WEC is not capable of absorbing any power: for high SL variations the translator might be stuck on the upper part of the generator hull and the buoy is submerged or could be resting on the lower end stop and the connection line to the buoy is slack.

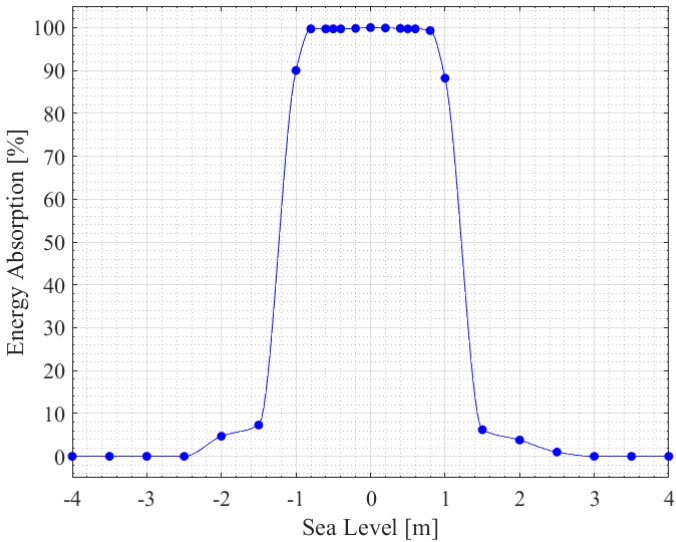

**Figure 8: Normalized annual energy absorption as a function of the SL for a L12 Uppsala WEC and for a sea state characterized by $H_s$ = 1 m and $T_e$ = 5 s. The markers indicate the results of the hydro-mechanic simulations, while the solid line serves as a guide to the eye. Adapted from Castellucci et al. (2016).**

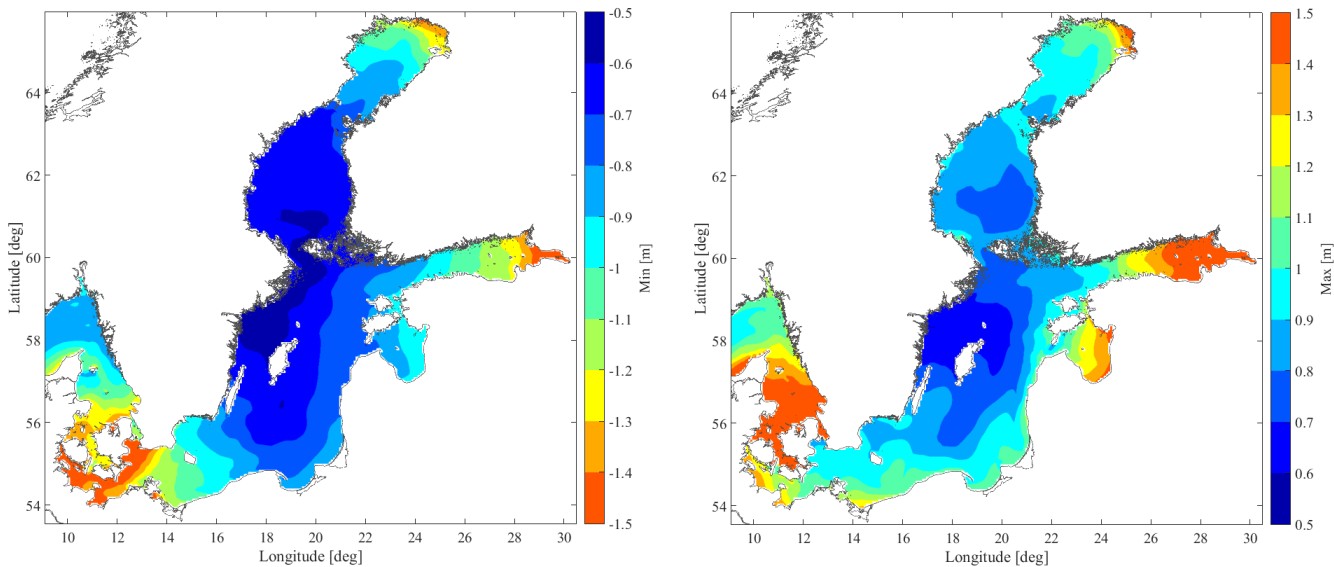

**Figure 9: Lowest minima (left) and highest maxima (right) of the SSH$_{1h}$ during the period 2007 to 2016, after subtracting the mean value.**

The validity of the results presented in Fig. 8 are limited to a specific sea state ($H_s$ = 1 m, $T_e$ = 5 s) and mostly dependent on the significant wave height, rather than the energy period (Castellucci et al., 2016). In particular, the plateau shown in Fig. 8 becomes wider with decreasing values of $H_s$. As a consequence, the energy absorption of WECs deployed in the patches of sea characterized by $H_s \leq 1$ m will be unaffected in the SL range of ±0.8 m at least. For the technology here considered, the MSSHR$_{10y}$ should be complemented with the minimum and maximum values of SSH: the WEC is not affected if the highest maximum and the lowest minimum do not exceed ±0.8 m at the desired site. The highest maxima and lowest minima in the studied area are shown in Fig. 9. For the purpose of the SWERM project, aiming at screening for suitable sites for wave energy utilization in the SEEZ, it is interesting to highlight areas with low enough SL variations to allow 100 % normalized annual wave energy absorption, as described by the case study and Fig. 8, with a typical wave climate for the SEEZ possibly interesting enough for energy conversion purposes. For this reason, we have generated a map of $H_s$ for ice-free conditions within the SEEZ, illustrated in Fig. 10. Ice-free conditions are more interesting for wave energy conversion purposes. These simulations are completely separate from the SL variations, but they use the same geographical grid network and spatial resolution.

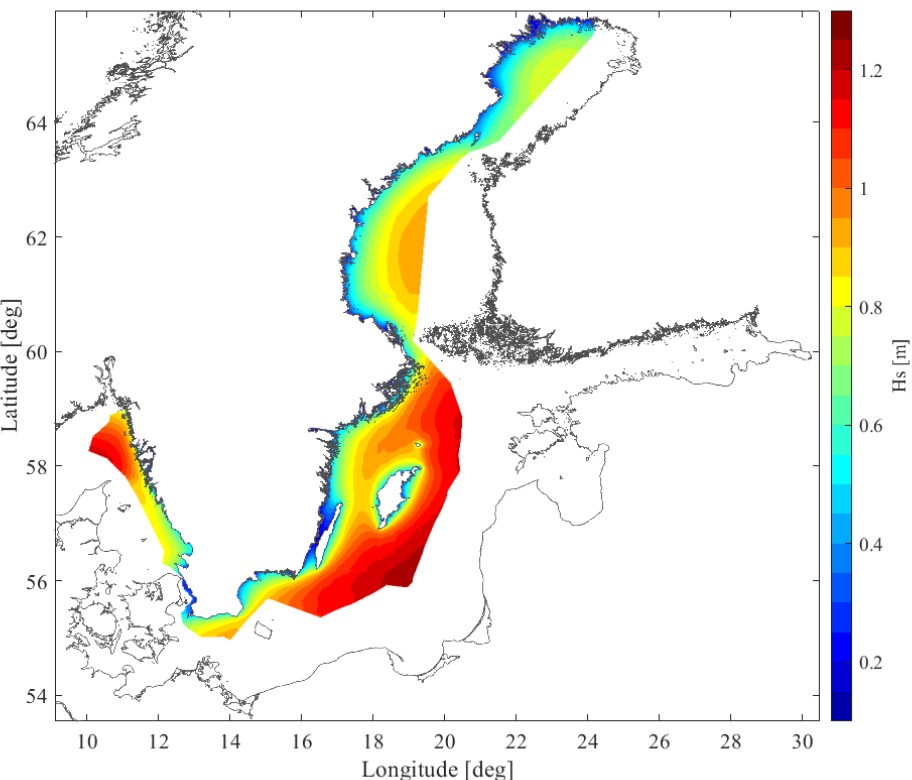

**Figure 10. Ice free average significant wave height, $H_s$, in the SEEZ from a 16 year high-resolution model simulation from the SWERM-project with methods described in (Strömstedt et al., 2017) and (Nilsson et al., 2019).**

$H_s$ has been estimated within the SWERM-project (Strömstedt et al., 2017), and methods for modelling and hindcasting are described in (Nilsson et al. 2019). In the wave climate modelling ice concentration below 30 % is considered ice-free. Above 30 % ice concentration, the sea is modelled as flat surface and energy is assumed to be completely attenuated by the ice (Tuomi et al., 2011). The percentage of time with ice concentration above 30 %, based on 35 years of ice data from 1980 to 2014, is mapped and presented in (Strömstedt et al., 2017). The difference in annual mean wave power estimates for ice-free conditions and ice-time-included statistics is mapped and presented by Nilsson et al. (2019).

For the purpose of illustrating the most interesting areas with regard to low SL variations and low negative impact on wave energy absorption the $MSSHR_{10y}$ presented in Fig. 5, is masked using the results in Fig. 9 and 10 as filters. The process of masking the range of SL with limiting values of maximum ($\leq$ +0.8 m), minimum ($\geq$ -0.8 m) and $H_s$ ($\leq$ 1 m) results in the left image of Fig. 11, which highligths the areas where the WEC energy absorption is unaffected by the changes in SL, i.e. part of the Northwestern and Eastern Gotland Basins, and a small area in the Bothnian Sea. The right image in Fig. 11 highlights areas where $H_s = 0.9 - 1.1$ m, corresponding with the $H_s$ that applies to the function in Fig. 8, and where the variations of the SL are less than ± 0.8 m and thus low enough to always allow a normalized energy absorption of 100 % based on a statistical confidence interval of 95 % defined by two standard deviations ($2SD_{10y} < 0.8$ m). A hypotetical WEC developer that is willing to pick a site where to deploy a park of Uppsala WECs may be interested to select one of the aforementioned basins with regard to SL variations.

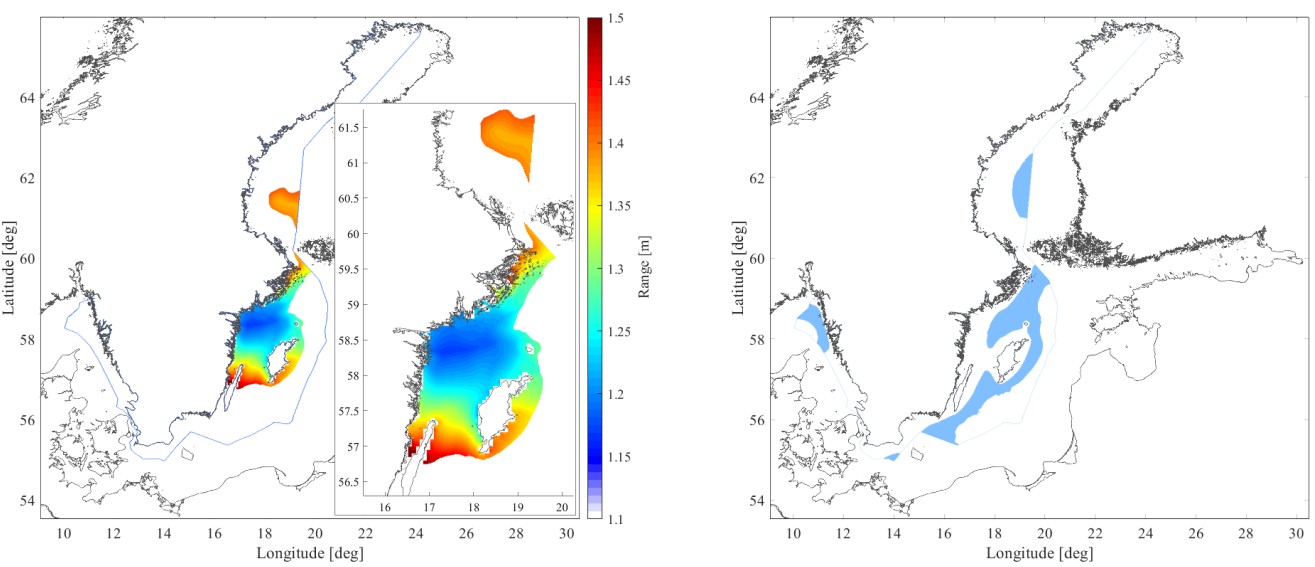

Figure 11: Left) Maximum range ($MSSHR_{10y}$) in areas with SL in the interval ±0.8 m, and significant wave height ≤ 1 m. The blue line indicates the boundary of the SEEZ. Right) The areas where $H_s$ is 0.9 – 1.1 m and where a normalized energy absorption with regard to SL is 100 % according to Fig. 8 with a confidence interval of 95%.

## 4 Discussion

When designing WECs and chosing suitable sites for wave parks deployment one generally has to consider wave power potential, water depth and seabed profile, distance to shore, accessibility and permissions, ice-concentration, SL variations, etc., which are all studied in the SWERM project for the SEEZ. This paper gives an overview of the SL variations in the SEEZ

and adjacent seawaters by means of the maps presented in Chapter 3. The same methodology described in Chapter 2 can be used to produce SL information layers (GIS layers) for other regions than the Baltic Sea.

As discussed among others by Johansson et al. (2001), Ekman (1996) and Stramska et al. (2013) the variability at a specific location of the Baltic Sea shows no apparent trend on a short time scale (10 days to 3 months), while it does on a seasonal time scale, when significantly higher variations in winter- compared to summer-time are observed. Moreover, they argue that the

spatial behaviour of the SD is clear on both interannual and seasonal time scales and it follows a specific pattern. These findings are in strong agreement with the results presented in this paper (see Fig. 6 and 7).

The highest decadal ranges presented in Fig. 5 show that the range of oscillations increases as we move out from the Northwestern Gotland Basin (Min value = 1.2 m) to the Bothnian Bay, the Danish straits and the Gulf of Finland (Max value = 4.3 m). The monthly ranges shown in Fig. 4 confirm the same spatial pattern and a not surprising seasonal tendency: the

range is lower during summer-time and higher during winter-time, in particular, July is the mildest month and January the one with the highest ranges.

The SD of the $SSH_{1h}$ confirms the same spatial and temporal patterns. Based on the $SD_{m,10y}$ (see Fig. 6), the most pronounced variability appears to occur during the winter-time (Nov-Jan), while the summer-time (May-Jul) is the one with the smallest variability. In general, the values of SD are quite large if compared with the rest of the globe, meaning that the variability of

the $SSH_{1h}$ is rather big. This has been shown as well by Ducet et al. (2000) in Plate 1 and by Thompson et al. (2016) in their Fig. 3. With reference to Fig. 7 in this study, the lowest $SD_{10y}$ values are found in the Bothnian Sea, Åland and Archipelago Sea, Gotland Basins, characterized by $SD_{10y} \leq 0.1$ m.

Note that a gap in the $SSH_{1h}$ data set has been identified during a few days in February 2008 and from the 24/2 to the 10/3 of 2012. This does not influence the results in a drastic way considering that February and March are not the most critical months

and that the missing data points are a small percentage (~ 0.5 %) of the total analysed data set. Regarding the peaks of $SSH_{1h}$ that are important when calculating the maximum ranges, the reanalysis model of SMHI tends to underestimate them. However, the correlation between model and observations is 0.91, and the RMS error is 9 cm for the Baltic Sea (Copernicus, 2018). An educated guess by SMHI would be that the underestimation is about 10 %. In general, the model responds correctly to changes in air pressure, winds, tides, and so on. The fluctuations of SL caused by barotropic salt water inflow events are

captured by the model, but do not affect drastically the maximum range. As an example, the major Baltic inflow event of December 2014 (Mohrholz, 2018) do not significantly influence the results neither in the Skagerrak nor in the central Baltic Sea, as illustrated in Fig. 3. In fact, as suggested by (Mohrholz, 2018), the majority of large inflow events are related to sea

level changes between 30 and 60 cm. In general, analysing the origins of the MSSHR was not in the scope of the study. Further investigation can be conducted as future work.

As mentioned before, low-frequency changes in SL may affect the performance of WECs. The case study presented in this paper aims to give an idea of the magnitude of the problem and to provide an example for WEC developers. A specific point absorber, the Uppsala WEC, and a representative annual average significant wave height ($H_s$) of 1 m are here considered. The first assumption limits the validity of the results for other devices: the energy absorption as a function of the SL variation (Fig. 8) should be carefully simulated or measured case by case. The second assumption reduces the scatter diagram of the sea state occurrences to one average state at an unspecified site: a WEC developer should select the most suitable sites on the basis of e.g. the accessibility and the wave power resource, then calculate the energy output for different sea states and aggregate the results in order to narrow down the number of suitable sites.

For the examined case, the areas where the WEC energy absorption is unaffected by the changes in SL are part of the Gotland Basins and a limited area of the Bothnian Sea, where the $MSSHR_{10y}$ is contained in the interval [1.15 - 1.55] m (see Fig. 11). If a more detailed analysis would be carried out, considering e.g. the full scatter diagram of sea states at each site, then the basins highlighted in Fig. 11 would certainly be different. Moreover, solutions for mitigating the negative effect of SL variations may be considered, e.g. the stroke length of the Uppsala WEC could be extended by applying changes in the design of the generator, or a compensation system to regulate the length of the connection line could be included in the design of the converter (Castellucci et al., 2016). Integrating a solution into the WEC design would increase the number of sites for wave park deployment, but most likely at higher capital investment cost.

Finally, it should be mentioned that according to the wave power technology one wants to investigate, a more detailed analysis of the frequency of occurrence of high ranges at a chosen site could be useful. This choice is dictated by the requirements set by every specific wave energy technology.

## 5 Conclusions

The dependency of the energy absorption on the low-frequency SL variation for wave energy converters is a matter of interest for different WEC technologies. For this reason, the changes in SL in the SEEZ and adjacent seawaters have been investigated in the frame of the SWERM project. The study carried out in this paper aims to give a deeper understanding of the variability of the SL in those basins and to provide an information layer (GIS layer) that, once the SWERM project will be completed, will be combined with other layers of information (GIS layers) to suggest suitable sites for wave park deployment.

From the calculation of the $SSH_{1h}$ standard deviation, it is clear that the variation of the high-frequency oscillations during the latest decade are limited especially in the Bothnian Sea, Åland and Archipelago Sea, Gotland Basins, where $SD_{10y} \leq 0.1$ m. The maximum range of these variations increases as we move out from the Northwestern Gotland Basin to the Bothnian Bay,

the Danish straits and the Gulf of Finland. The $MSSHR_{10y}$ varies from the lowest value of 1.2 m (Northwesten Gotland Basin) to the maximum value of 4.3 (Gulf of Finland) during the period 2007-2016. The seasonal variability is evident: it is more pronunced during the winter-time and less during the summer-time. The spatial variability is also noticeable and almost independent of the month: the highest oscillations are found in the Bothnian Bay, the Gulf of Finland, the Kattegat and in the

Danish straits, reaching up to 4 m in the Gulf of Finland. More constant conditions are found in the Northwestern Gotland Basin, characterized by $MSSHR_{10y}$ of 1.2 to 1.5 m, with very low range during summer-time (< 0.7 m).

With the purpose of comprehending how the SL can affect a point absorber WEC, an example has been shown. An Uppsala WEC with specified features has been considered and the energy absorption as a function of the SL has been evaluated, assuming a wave climate of relevance for wave energy conversion with a high rate of occurrency in the SEEZ and adjacent

seawaters. From a $MSSHR_{10y}$-point-of-view, areas suitable for deployment are found in the Bothnian Sea, Northwestern and Eastern Gotland Basins, where the 10-year maximum range is contained in the interval [1.15 − 1.55] m.

The data sets here displayed by means of geographic maps will be available on-line or on request by the end of the SWERM project, and can be used by WEC developers to perform analysis according to the technology and models they work with. Moreover, the data will be used to complete the SWERM project that intends to merge different layers of ocean data (GIS

layers) for the SEEZ. Further information on where to retrieve layers and/or data sets will be available on the following homepage in the fall of year 2020: http://www.teknik.uu.se/electricity/research-areas/wave-power/.

**Acknowledgments**

The authors would like to thank The Swedish Energy Agency for funding the project (Project no. 42256-1) within the national

Swedish research program for marine energy conversion. The project is also supported by the Swedish STandUP for Energy research alliance, a collaboration initiative financed by the Swedish government. StandUP for Energy is acknowledged for providing a research infrastructure. The authors would like to thank the Swedish Meteorological and Hydrological Institute (SMHI) for the geological input data on sea level variations from the Copernicus project and, in particular, Dr. Lars Axell for valuable input on the simulations performed by SMHI. The authors would also like to thank Dr. Erik Nilsson at the Department

of Earth Sciences, Uppsala University, for the average ice-free significant wave height data in Fig. 10.

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
