# Peer review of "Sea Level Variability in the Swedish Exclusive Economic Zone and adjacent seawaters: Influence on a Point Absorbing Wave Energy Converter"

_Ocean Science, 2019_

## Referee Comment (RC1) · Anonymous Referee #1 · 28 May 2019

The article "Sea Level Variability in the Swedish Exclusive Economic Zone and adjacent seawaters: Influence on a Point Absorbing Wave Energy Converter" investigates sea level variability in the Baltic Sea to be used for the identification of a location to deploy wave energy converter (WEC) systems. This study if of great interest for WEC developers, however, before I can recommend publication, I recommend a major revision.

Major comments:

[Figure]

The main result of this study is shown in Figure 10b, defining the optimal sites for the deployment of WECs when only considering MSL variability. However, as written by the authors on page 15, line 5-6, this might change when considering for example the full sea states at each side. As the motivation of this paper is to provide a tool for WEC developers for choosing deployment sides, who need to take not only the MSL variability into account, I would like to suggest to extend Figure 10, showing also a figure on how the regions of optimal deployment side would change when not only MSL variability is considered.

Regarding sea-ice, on page 14, line 1-2 it is stated that the ice-concentrations have to be considered, but in this study it is only computed for ice-free conditions as all ice-variables are assumed to be zero (page 5, line 1). In addition, on page 13, line 2-6 (result section) was written an overview about ice-concentration and wave power summarized from other studies. May one could move this paragraph to the introduction and than include an argumentation why ice can be neglected in this study in the method section near page 5, line 1. And what happens with the WEC when sea-ice exists?

Page 14, line 28-32 and page 15 line 17-18: How are the other WECs systems, mentioned in the introduction, are influenced by MSL variations compared to the Uppsala WEC?

Please include equations to the manuscript for the calculations performed in this study. Although the calculations are not too complex, it would be much easier to grasp what has been calculated (and from which data) from the additional equations than only from text.

In addition, a time period from 2007-2016 was chosen, which includes a strong barotropic salt water inflow event in December 2014 (driven by a strong sea level gradient between the Baltic Sea and the Kattegat). What influence has this event on the MSL variability and the WEC systems? What about other short term variability? Please also discuss this or why it is neglected. The chosen time period from 2007-2016 represents the current situation. How are possible future changes included in this tool, as this is also needed for the decision making process of the WEC developers?

Minor comments

Figure 1: This figure is very similar to the figure in Castellucci et al. (2016). Therefore, please include something like "adapted from ... " in the figure caption as reference.

Page 2, line 22: "to give an example, let us consider..." → The Uppsala WEC system is considered as example.

Page 4, line 13 - page 5, line 12: Unfortunately, the paragraph is not clear to me. Are the simulations done within this study or are they performed by SMHI in another study? Is the MSL extracted from NAMOD with a 44 km grid resolution or is it the one available at marine.copernicus.eu? Or is the second one used for validation? Is the 44 km grid resolution not a bit coarse for studies in the Baltic Sea? Is the validation of the model results part of this or a different study? etc. Please rewrite this section to clarify what simulations performed and what data has been used within this study. Please also acknowledge all work or data provided from others (if this is the case) with references and mention them in the acknowledgment section.

Page 5, line 2: Please include the reference for the atmospheric forcing (HIRLAM data set)

Page 5, line 3: Modeled MSL is normally strongly affected by data assimilation. Therefore it would be very beneficial to have a short overview on which variables are assimilated and which are affected and how, before referring to the studies describing the assimilation process in detail.

Page 5, line 12: Why are extreme events are neglected in this study. What impacts have extreme events on WECs?

Page 5, line 26: Is the MSL range computed from the hourly data?

Page 6, line 15: semester → time period

Page 8, line 9: When you computed the pooled standard deviation, did you include a weighting. If yes, please specify it.

Figure 3 and 4: Please use the same color scale to be able to compare the figures.

Figure 5 and 6: Please use the same color scale to be able to compare the figures.

Page 10, line 8-11 and Figure 7: please clarify if this is from Castellucci et al. (2016) or results of this study. Please include light gray grid lines for easier identification of the $\pm 0.8$m MSL.

Page 12, line 8: "It is interesting to filter out areas with low enough MSL ...", Did you mean that you wanted to "filter out" regions with higher MSL variations and keep the areas with low enough MSL variations, here? Or did I misunderstood something?

Page 16, line 4: "are available on-line", please add where to find them

Please discuss the possibility of using this study to find deployment location in other regions as the Baltic Sea in the discussion section

References: please include the DOIs for all the references

———————————————

---

## Referee Comment (RC2) · Anonymous Referee #2 · 30 May 2019

This paper presents an analysis of sea level variability in the Baltic Sea with its potential impact on the energy yields of some wave energy converter devices. The structure of the paper is good and the results are clearly shown with some novel information.

The main question that I am left with though is with regards to the wave energy component. Every site chosen for energy generation is determined by cost of energy, and this in turn is determined by the energy yield and cost of installation and maintenance. How much energy is actually available for the wave device in the region with a constant sea level and how much does the change in sea level affect this? I can't get a feel for how

much of an impact the change in sea level has from the data shown here. I don't even have a clear idea of the frequency of these large deviation sea level periods. Perhaps it would be useful to have a histogram of the sea levels at the representative location. It would be helpful if instead of a single design wave, the energy from a device across a representative wave field was shown and how this is affected by the changing sea level. The second related area, is that I am left questioning if the sea level changes have a major negative impact on the energy yield why use this type of wave device? You suggest that there are fixes to these problems in your discussion but why not just include these solution in all models? Presumably because of the cost? Some description of this would be helpful in providing a basis for justifying this particular energy device.

A more minor point is with regards to use of mean sea level. As far as I can determine you actually use the sea level as an hourly mean of your time series. This is obviously right for this context as the waves of interest are on much shorter timescales. However, when I read MSL I think of much longer term sea level. The tides are measured around MSL but this obviously doesn't change on an hourly basis, and the use of the terms in sea level rise is also obviously in a much longer context. It would be helpful to indicate the exact MSL you mean in your paper. Sometimes you use the MSL1h and others just MSL and the two are not generally interchangeable in a wider oceanic context.

Some typos

In the abstract on line 9 "linear systems with at a limited..." should remove the at

On page 2 and line 18 the final element in the list "pole tide" should have and the added before.

On page 5 the last word Ensemble is spelt incorrectly on line 2

On line 11 change the word have to has

On line 13 at the end of the line I think it should be sea level variations

On line 20 insert it into "one may find it preferable"

On line 33 where talking about the weight did you mean except the buoy rather than but the buoy?

On page 10 the second sentence of the Case Study section needs to be rewritten to make it clearer

On page 14 the addition of a in "the energy absorption as a function..." in line 31

On page 15 the word than should be then in line 1

At the end of line 20 I think that wave park should be singular

On line 20 an a should be inserted in "energy absorption as a function..."

---

## Author Comment (AC1) · 25 Jun 2019

The article "Sea Level Variability in the Swedish Exclusive Economic Zone and adjacent seawaters: Influence on a Point Absorbing Wave Energy Converter" investigates sea level variability in the Baltic Sea to be used for the identification of a location to deploy wave energy converter (WEC) systems. This study if of great interest for WEC developers, however, before I can recommend publication, I recommend a major revision.

Major comments:

[Figure]

[Figure]
 The main result of this study is shown in Figure 10b, defining the optimal sites for the deployment of WECs when only considering MSL variability. However, as written by the authors on page 15, line 5-6, this might change when considering for example the full sea states at each side. As the motivation of this paper is to provide a tool for WEC developers for choosing deployment sides, who need to take not only the MSL variability into account, I would like to suggest to extend Figure 10, showing also a figure on how the regions of optimal deployment side would change when not only MSL variability is considered.

[Figure]
 Regarding sea-ice, on page 14, line 1-2 it is stated that the ice-concentrations have to be considered, but in this study it is only computed for ice-free conditions as all ice-variables are assumed to be zero (page 5, line 1). In addition, on page 13, line 2-6 (result section) was written an overview about ice-concentration and wave power summarized from other studies. May one could move this paragraph to the introduction and than include an argumentation why ice can be neglected in this study in the method section near page 5, line 1. And what happens with the WEC when sea-ice exists?

[Figure]
 Page 14, line 28-32 and page 15 line 17-18: How are the other WECs systems, mentioned in the introduction, are influenced by MSL variations compared to the Uppsala WEC?

[Figure]
 Please include equations to the manuscript for the calculations performed in this study. Although the calculations are not too complex, it would be much easier to grasp what has been calculated (and from which data) from the additional equations than only from text.

[Figure]
 In addition, a time period from 2007-2016 was chosen, which includes a strong barotropic salt water inflow event in December 2014 (driven by a strong sea level gradient between the Baltic Sea and the Kattegat). What influence has this event on the MSL variability and the WEC systems? What about other short term variability? Please also discuss this or why it is neglected. The chosen time period from 2007-2016 rep-

resents the current situation. How are possible future changes included in this tool, as this is also needed for the decision making process of the WEC developers?

Minor comments

[Figure]
Figure 1: This figure is very similar to the figure in Castellucci et al. (2016). Therefore, please include something like "adapted from ... " in the figure caption as reference.

[Figure]
Page 2, line 22: "to give an example, let us consider..." → The Uppsala WEC system is considered as example.

Page 4, line 13 - page 5, line 12: Unfortunately, the paragraph is not clear to me. Are the simulations done within this study or are they performed by SMHI in another study? Is the MSL extracted from NAMOD with a 44 km grid resolution or is it the one available at marine.copernicus.eu? Or is the second one used for validation? Is the 44 km grid resolution not a bit coarse for studies in the Baltic Sea? Is the validation of the model results part of this or a different study? etc. Please rewrite this section to clarify what simulations performed and what data has been used within this study. Please also acknowledge all work or data provided from others (if this is the case) with references and mention them in the acknowledgment section.

Page 5, line 2: Please include the reference for the atmospheric forcing (HIRLAM data set)

Page 5, line 3: Modeled MSL is normally strongly affected by data assimilation. Therefore it would be very beneficial to have a short overview on which variables are assimilated and which are affected and how, before referring to the studies describing the assimilation process in detail.

[Figure]
Page 5, line 12: Why are extreme events are neglected in this study. What impacts have extreme events on WECs?

Page 5, line 26: Is the MSL range computed from the hourly data?

[Figure]

Page 6, line 15: semester → time period

Page 8, line 9: When you computed the pooled standard deviation, did you include a weighting. If yes, please specify it.

Figure 3 and 4: Please use the same color scale to be able to compare the figures.

Figure 5 and 6: Please use the same color scale to be able to compare the figures.

Page 10, line 8-11 and Figure 7: please clarify if this is from Castellucci et al. (2016) or results of this study. Please include light gray grid lines for easier identification of the ±0.8m MSL.

Page 12, line 8: "It is interesting to filter out areas with low enough MSL ...", Did you mean that you wanted to "filter out" regions with higher MSL variations and keep the areas with low enough MSL variations, here? Or did I misunderstood something?

Page 16, line 4: "are available on-line", please add where to find them

Please discuss the possibility of using this study to find deployment location in other regions as the Baltic Sea in the discussion section

References: please include the DOIs for all the references

[revised manuscript text omitted]

---

## Author Response (AR2)

Dear Editor,

We have added the information You have requested and implemented the changes highlighted below with track changes.

In particular, we have modified Figure 2 and included Figure 3 to shows the time series of SL at two representative locations. Moreover, we have commented on that within the Discussion chapter in order to clearly answer the Reviewer 1 question regarding the salt water inflow event during December 2014.

We hope that this new version of the manuscript will satisfy You and that the paper will be accepted for publication in Your Journal. We look forward to hear from You.

Best regards,

Valeria Castellucci

Corresponding Author

[revised manuscript text omitted]

---

## Author Response (AR3)

Dear Editor,

We have understood that the Reviewer 1 seems still concerned whether or not major inflow events are included in the dataset generated by the model simulations in the Copernicus project. The Swedish Meteorological and Hydrological Institute, that provided the simulated data to the Copernicus project, confirmed that the major inflows are included in the data set we have used for the analysis presented in this study.

The figure below shows observations and simulated data at Landsort for the period 1st November 2014 to 31st January 2015, which includes the December inflow event.

[Figure]

In general, analysing the origins of the MSSHR was not in the scope of the study. We have taken the best available dataset and analysed it from a wave energy perspective for the purpose of finding suitable conversion sites from the sole standpoint of the SSH variability. However, in the discussion, we have added a statement regarding possible future work as requested by the Reviewer 1.

We hope that this new version of the manuscript will satisfy You and the Reviewer 1, and that the paper will be accepted for publication in Your Journal.

Best regards,

Valeria Castellucci

Corresponding Author

[revised manuscript text omitted]